# Participants’ Perspective of Engaging in a Gym-Based Health Service Delivered Secondary Stroke Prevention Program after TIA or Mild Stroke

**DOI:** 10.3390/ijerph182111448

**Published:** 2021-10-30

**Authors:** Maria Sammut, Kirsti Haracz, Coralie English, David Shakespeare, Gary Crowfoot, Michael Nilsson, Heidi Janssen

**Affiliations:** 1School of Health Sciences, College of Health, Medicine and Wellbeing, University of New-Castle, Callaghan, NSW 2308, Australia; Kirsti.haracz@newcastle.edu.au (K.H.); coralie.english@newcastle.edu.au (C.E.); david.shakespeare@newcastle.edu.au (D.S.); michael.nilsson@newcastle.edu.au (M.N.); heidi.janssen@health.nsw.gov.au (H.J.); 2Priority Research Centre for Stroke and Brain Injury, Hunter Medical Research Institute, New Lambton Heights, NSW 2308, Australia; 3School of Nursing and Midwifery, College of Health, Medicine and Wellbeing, University of Newcastle, Callaghan, NSW 2308, Australia; gary.crowfoot@newcastle.edu.au; 4Centre for Rehab Innovations, College of Health, Medicine and Wellbeing, University of Newcastle, Callaghan, NSW 2308, Australia; 5Hunter New England Local Health District, Community and Aged Care Services Community Stroke Team, New Lambton Heights, NSW 2308, Australia

**Keywords:** transient ischemic attack, TIA, mild stroke, secondary stroke prevention, physical activity, community health, qualitative descriptive study

## Abstract

People who have had a transient ischemic attack (TIA) or mild stroke have a high risk of recurrent stroke. Secondary prevention programs providing support for meeting physical activity recommendations may reduce this risk. Most evidence for the feasibility and effectiveness of secondary stroke prevention arises from programs developed and tested in research institute settings with limited evidence for the acceptability of programs in ‘real world’ community settings. This qualitative descriptive study explored perceptions of participation in a secondary stroke prevention program (delivered by a community-based multidisciplinary health service team within a community gym) by adults with TIA or mild stroke. Data gathered via phone-based semi-structured interviews midway through the program, and at the end of the program, were analyzed using constructivist grounded theory methods. A total of 51 interviews from 30 participants produced two concepts. The first concept, “What it offered me”, describes critical elements that shape participants’ experience of the program. The second concept, “What I got out of it” describes perceived benefits of program participation. Participants perceived that experiences with peers in a health professional-led group program, held within a community-based gym, supported their goal of changing behaviour. Including these elements during the development of health service strategies to reduce recurrent stroke risk may strengthen program acceptability and subsequent effectiveness.

## 1. Introduction

A transient ischemic attack (TIA) is a focal and rapid neurological disturbance resulting from a temporary disruption of arterial blood flow with no residual brain tissue death (infarction) and complete resolution of symptoms [1,2]. While persistent disruption of blood flow results in brain tissue infarction and subsequent stroke, people with mild stroke generally experience rapidly resolving symptoms and are assessed as having minimal to no residual physical deficits [3,4]. There is an assumption that individuals return to pre-stroke event activity levels following a TIA or mild stroke and require little to no referral to follow up services [5,6]. However, this patient population face a high long-term risk of recurrent stroke [7]. 

Physical inactivity is a significant independent modifiable risk factor for recurrent stroke and TIA [8]. Recent evidence using measured accelerometer outcomes shows recurrent stroke is more likely among people with TIA and mild stroke who engage in low levels of moderate to vigorous physical activity (MVPA) and have high levels of visceral adiposity [9]. Regular MVPA is associated with cardiorespiratory benefits of improvements in blood pressure [10,11], blood lipids [12] and vascular endothelial function [13]. Evidence-based guidelines encourage people with TIA or mild stroke to engage in regular MVPA for 150 min or more per week as a strategy to help ameliorate the risk of recurrent stroke [14,15,16]. However, people with TIA and mild stroke are less active than peers who have not had a similar stroke event [17].

Evidence shows that secondary stroke prevention programs explicitly targeting physical activity are feasible, safe, and effective in facilitating favourable changes in modifiable lifestyle behaviours within the research institute setting [10,18,19]. However, there is limited evaluation of community-based secondary stroke prevention programs delivered by health services that support people with TIA and mild stroke to change physical activity behaviours [20,21]. Understanding people’s experience of participating in health service delivered secondary stroke prevention is vital to ensure that similar programs implemented in ‘real world’ settings meet the complex individual needs, provide good support for reducing stroke risk factors, and facilitate participation [22]. This study aimed to explore the perceptions and experiences of people with TIA or mild stroke who participated in a secondary stroke prevention program delivered by a health service in a community-based gym setting. 

## 2. Materials and Methods

### 2.1. Study Design

The consolidated criteria for reporting qualitative research (COREQ) guided the reporting of this study [23]. The methodology of qualitative description was carried out from a constructivist perspective, with data gathered using semi-structured interviews and analysed using grounded theory methods [24,25]. Qualitative description is a methodology described by Sandelowski (2000) [26] which is well suited to health service research when investigating the how and what questions associated with patient attitude and experience [25,26]. The nature of our research question determined a methodological approach, not to generate a conceptualised theoretical explanation of behaviour, but to provide a rich and detailed description of participant experience [25,27]. However, qualitative description does not have specific data analysis strategies [26]. We decided to utilise some of the methods associated with Charmaz’s constructivist grounded theory analysis, as they provide a delineated process for data analysis [24]. Furthermore, the constructivist approach acknowledges that analysis outcomes are a co-creation of participant experience and researcher interpretation [24].

### 2.2. Context

This study was embedded within a non-randomised, controlled, health service evaluation trial, Service change and Supporting Lifestyle and Activity Modification after TIA (S+SLAM-TIA) [21], where SLAM-TIA refers to the secondary stroke prevention program developed and delivered by the Community Stroke Team. The Community Stroke Team is a health service available to the people living within a large regional city in the Hunter region, New South Wales [21]. It aims to support adults with TIA or mild stroke develop skills necessary for lifestyle modifications, inclusive of independent MVPA at guideline recommendations [21]. Participants receive education on strategies to modify lifestyle risk factors for recurrent stroke, and participate in an individualised routine of aerobic and resistance exercises, within a group setting, under health professional guidance for six weeks in a community gym. Following this, participants receive a further 12 weeks of fortnightly telehealth-based health coaching [21]. Figure 1 illustrates the trial outline. Eligibility to participate in the program depends on referral to the health service following a diagnosed TIA or mild stroke; the participant must have little to no physical impairment (NIHSS ≤ 3, mRS 0-1) and be able to walk a flight of stairs [28].

### 2.3. Recruitment

All participants who had provided written and informed consent to participate in the overarching trial were invited to this study and contacted via telephone one week before the end of the face-to-face component of the trial, and one week before the end of the telehealth-based sessions by MS to confirm consent and organise a convenient time for the interview. All participants were aware they were not obliged to agree to an interview, answer questions if they chose not to, and could exit the study at any time without jeopardising treatment. Table 1 lists participant characteristics.

### 2.4. Sample

Time constraints associated with the overarching trial influenced sampling with all consenting participants interviewed. Therefore, the sampling of this study was from the pool of completed interviews collected between October 2018 and September 2020. First, we purposively sampled twelve interviews for open initial coding using a maximum variation approach representing the broadest range of participant characteristics. Secondly, during focused coding, interviews were consecutively sampled from the pool.

### 2.5. Generating Data

Data was collected via one-to-one, telephone-based, semi-structured interviews (scheduled for participant convenience), and recorded using a digital voice recorder (Olympus WS-825, (Olympus Corporation, Tokyo, Japan)). Two authors (one female (MS), one male (DS), both qualified physiotherapists), independent of the intervention trial, received training in qualitative interviewing techniques and conducted the interviews. Participants were usually at home at the time of the call, and participants could choose to have a support person present during the interview. Interviews occurred at the end of (i) the education and exercise component of the secondary prevention program (week 6) and (ii) the telehealth-based health coaching (week 18) (see Figure 1). Interviews lasted between 30 and 50 min.

An interview guide (see Appendix A), developed from findings of a pilot study investigating people’s readiness for secondary stroke prevention [29], allowed interviewers the flexibility of following topic points while maintaining the flow of conversation and allowing for an extension of discussion. Field notes written during the interviews reflected on salient points (e.g., tone of voice, emphasised points, expression of emotion). A unique code anonymised the interviews. MS transcribed all interviews verbatim. Data was stored on a secure University of Newcastle database and secured file drive.

### 2.6. Data Analysis

The first author (MS) conducted inductive data analysis using grounded theory procedures associated with initial open coding and focused coding [24]. The second author (KH), a qualified occupational therapist, experienced in qualitative research, provided a weekly review of the analysis process focusing on how well the codes accounted for the data, the appropriateness and coherence of emergent codes to describe the data, distinction between the codes, and clarity of relationships between codes. Qualitative research software (Nvivo 12, QSR International Pty Ltd, Doncaster, Australia) [30] was used throughout the coding process to create nodes, organise categories, and provide an audit trail. Interviews from both collection points were analysed concurrently. The process of open initial coding generated a large number of codes. Codes were grouped by similarity to form preliminary, tentative categories. Discussion between MS, KH and two additional authors (HJ and CE, both experienced physiotherapists and specialised in stroke research) added further insight into the analysis process for a richer interpretation of the data. Focused coding commenced once patterns were observed in the codes. Continuous comparative analysis of data from newly coded interviews to existing data continued beyond the point of sufficiency, ensured concept categories were well developed, and offered a rich insight into participant perceptions. (See Appendix A Example of coding tree). All analysis team members discussed and agreed upon the final concepts, associated categories, subcategories, and dimensions. Memo writing occurred throughout the analysis process to summarise key points, note issues raised during analysis, and capture ideas that required further development. The inclusion of negative responses ensured that contradictory factors were subject to further analysis. Quotes presented by pseudonym and age illustrate the sentiment of the majority of participants for the result discussed.

### 2.7. Trustworthiness

Many mechanisms were used to support study trustworthiness [31]. Member checking was not possible because of potential bias in the overarching trial. However, regular oral, analytical discussion and peer critique ensured strong logic and data linkage to emergent coding and categories to strengthen study credibility [31]. Analysis of all interviews ensured a wide range of participant perceptions and various experiences. Furthermore, input from experienced researchers within the field of stroke and qualitative research contributed to the quality of analysis to ensure data interpretation occurred in context. Journaling study logistics, using a standardised interview guide, maintaining a methods log, and detailed memo writing, to augment contemplation and provide consistency between data and findings, strengthened study dependability and audibility. Ongoing scrutiny of personal biases, experience, and background influences (reflexivity) continued throughout the analysis process to reflect on any researcher preconceptions that may influence the interpretation of participant voice [24].

## 3. Results

### 3.1. Results

Thirty participants (43% female, 36% TIA, mean age 66 (standard deviation 10) years, range 42 to 86 years) provided 51 interviews for data analysis (20 participants agreed to interview at both time points); 85% were overweight or obese (BMI > 25 kg/m^2^), and most had their stroke event less than six months before entry into the program.

### 3.2. Findings

The analysis produced two concepts: “What it offered me” and “What I got out of it”. Table 2 lists these concepts with corresponding categories and subcategories. “What it offered me” describes aspects of the program that shaped participant experience. “What I got out of it” describes the perceived benefits of program participation.

### 3.3. What It Offered Me

#### 3.3.1. Health Professional Support

The majority of participants believed that health professional support was central to the program.

“Well, she was the program…. Without her, there was no program. She made it all happen…kept everybody enthusiastic and when she wasn’t there…nobody cared” Geoff 69yrs.

In person

Most participants thought in-person health professional support was essential during the first few weeks of the program. Many participants were happy to have less in-person interaction towards the end of the intervention; however, some preferred continued support as they found it challenging to maintain enough motivation or were anxious about engaging in independent MVPA.

“I don’t think by the end we needed more, I certainly wouldn’t have liked less, it’s better to have someone there, that’s most important I think because there’s a little bit of hesitation–you know–what do I do next”, Anne 76yrs.

The approachability of the health professional was an essential aspect of the program as participants valued willingness to answer questions, listen, and an expression of empathy. 

“This lady would actually listen to me…...she took an interest”, Dean 42 yrs.

Participants valued having the health professional standing alongside them and showing them what to do while engaging in MVPA, for guidance and feedback. Reassurance that they were performing their prescribed exercise routine correctly and safely was necessary for most.

“It was good to have them there and that, showing us what to do and everything”, Bob 76yrs.

Some participants led sedentary lifestyles before their stroke event, with little to no experience of engaging in physical activity within a gym setting, and were anxious that increasing their heart rate would bring on a stroke. Receiving in-person encouragement helped them feel safer about engaging in MVPA.

“They were very understanding about that and they put no pressure on me or no guilt or ‘you should or shouldn’t’. They were very accepting”, Roger 76yrs.

While encouragement to extend themselves and try new things was valuable, some participants highlighted the importance of respecting individual choice.

“You’re not pushed to do some something that you’re really probably not up to do”, Linda 59yrs. 

Having the health professional in person monitoring them was significant for most participants. Monitoring heart rate and blood pressure while engaging in MVPA gave many reassurances and a sense of safety.

“They spent plenty of time with us to make sure we could do the exercise and that we were ok. They were constantly coming over check to see how your heart rate and breathing was going. You were always monitored so I didn’t have any problems”, George 67yrs.

Helpful information

Participants wanted helpful information and spoke positively about the information provided in the program, whether this was new knowledge for them or reinforcing what they already knew.

“When I went in to the program I had no idea of the information that I needed, after being exposed to the education there were a lot of aha experiences”, Frank 62 yrs.

Health professional guidance in understanding the information during the education sessions contributed to information usefulness.

“The education was a little bit scary as to what can happen to you in the future…it made me more aware of trying to avoid having another episode”, Frances 65yrs.

However, some believed additional, one-to-one discussion would have helped identify personal knowledge gaps for tailoring the information to their needs.

“I would like….somebody to tell me all those things again and what you actually mean by them”, Roger 76yrs.

Participants generally appreciated being included in the conversation and considered the informal guided approach to learning more beneficial than didactic instruction.

“They spoke to people, not down at people or about people; everything was put into a conversation where you could relate to it”, Joe 63yrs.

Participants who wanted to progress their program independently thought that being part of the discussion allowed them to have queries answered ‘on the spot’ and engage in in-depth discussion if necessary.

“You needed them there to ask questions and practice the advice…I wanted to know, like where should my heart rate be”, Doug 63yrs.

Dealing with stroke-related psychological changes was unsettling for some. Some participants valued emotional support, open discussion, and education in helping them understand their anxiety and depressive states. However, others would have preferred further education on psychological changes associated with their TIA or mild stroke.

“A big part of my stroke…getting to terms with the emotional side and I don’t think that was dealt with”, Joe 63yrs.

Although the written material was well received, some participants preferred it collated as a formal journal for ease of use and tracking personal progress.

“I use the papers as a refresher to keep me on the right track, but it would be easier to use with a content”, Tony 59yrs.

Telehealth-based health coaching

Most participants believed the telehealth-based health coaching was a valuable adjunct to the program. Many thought that telephone contact helped keep them on track by reflecting on goals, discussing other issues of concern, identifying the need for further help, referral to supporting services, or receiving additional information.

“It just kept you a little bit topped up, or a wake-up call, they didn’t need to be long, but just enough for my concerns”, Dean 42yrs.

Participants like the personalised aspect of telehealth-based health coaching. Although the amount of time spent on the phone was not essential, informality, consistency of health professional support, and asking what they needed were vital factors for feeling included and valued. However, not all participants found the phone calls to be of use. Some found the sessions annoying, spoke of feeling disinterested, and tended not to continue phone contact.

“I needed it to be more informal, rather than direct questions, I wanted someone to say ‘how are you going today?’ you know “how are you feeling today? Take an interest in me as me”, Geoff 69yrs

#### 3.3.2. Being in a Group

Being in a group was the second category of “What it offered me”. Participants were generally positive about the group setting, even those who did not consider themselves the type of people who enjoyed group activities, or who were apprehensive about being in a group. 

“I’m not a group person to be quite honest, but I found it very easy with those people. The people make a huge difference”, Anne 76yrs.

Mutual support

Being part of a group gave some participants a sense of mutual support. In addition, being comfortable within the group was important for many who believed that it added an element of social interaction that encouraged companionship.

“I find that a lot of older people who have had stroke problems are missing out on getting a lot of talk with other people other than a neighbor”, Mike 73yrs.

Encouragement to share experiences was a program attribute that helped participants understand their stroke event by asking questions with others. However, it was acknowledged that sometimes in the group setting dominant people directed the flow of the conversation. 

“A big part was for me was being around people that have been through that experience. It was great to have a shared experience. Knowing I’m not alone -it’s probably therapy in itself”, Scott 54yrs. 

Realizing others had similar experiences was necessary for some participants, reassuring them that they were not alone, as some thought that their family and friends, although caring, could not comprehend their emotional struggles.

”Sometimes it’s a lonely path because you don’t feel that anybody relates to you. They don’t understand. But if you’ve got a group of people that you’ve met and they understand that they’re going through the same stuff, the same joys and tribulations, then you tend to affiliate and tend to reach out”, Terry 57yrs.

Mutual support brought a sense of accountability that gave some a sense that others relied on them to turn up, which provided the impetus to continue with the program.

“When you’re in a group, you are more likely to turn up because you have people relying on you. But when you are on your own you can always find an excuse to not go”, Linda 59yrs.

Although family members were encouraged to participate in the program to facilitate their understanding of the importance of recurrent stroke prevention, some participants believed there was a need for specific support sessions for family and carers, allowing them the opportunity to share experiences with other families/carers.

“Maybe something for the partners, people that are supporting the people with the stroke, maybe a separate session for them”, Scott 54yrs.

Exercising with others

A significant focus of being in a group was in the context of exercising with others, as many commented that the informality of the group setting offered them the ability to engage in their routine independently alongside others. 

“I did far better in a group than if I had to be by myself. We weren’t competitive but when we worked side by side on the machines there was a bit of you know “what level are you on, how long have you been on”, we could pace ourselves by being together”, Phil 73yrs.

Many enjoyed the social interaction of having others beside them, the opportunity for companionship and friendly banter while exercising together. 

“They were a good group of people and we were sort of helping each other. It was what we were doing and what we got out of it”, Doug 63yrs. 

Observing others achieve and having someone in the gym with similar needs and goals enabled some participants to pace themselves and inspired them to persevere with individualized programs.

“It was good to have them there, someone in the same boat as you; it was good to know that they were striving for what you were striving for”, Mike 73yrs.

#### 3.3.3. Meeting My Needs

The third and final category of “What it offered me” was “Meeting my needs”, which included two subcategories: experiencing MVPA and convenience, which broadly account for the program meeting participant individual needs. However, some participants were apprehensive about aggravating pre-existing comorbidities and thought in-depth discussion with the health professional would help address individual needs.

“That was my overall concern … individually profiling the person and establishing what individual needs they have or think that they need to be working towards … specifically”, Terry 57yrs.

Experience MVPA

Using a community gym was a new experience for some participants who were anxious about their ability to engage in their prescribed program or concerned about the safety of engaging in MVPA, believing that it may cause a subsequent stroke.

“At the beginning I wasn’t quite sure how I would go…..I was worried…. a little whether, firstly I could do it or whether I could last the distance and whether the body could stand up to it”, Anne 76yrs.

One of the aspects of experiencing MPVA was understanding intensity.

“The gym sessions were really important to me, because I didn’t know how far to push it”, Doug 63 yrs.

Most participants believed their individualized routine was set at an intensity harder than they anticipated. Generally, however, they reported that experiencing MVPA in the community gym setting enabled them to see that they could engage in physical activity at this intensity. In addition, participants believed that the opportunity to experience MVPA helped with knowing what to do. Many felt safe that the health professional had set upper limits and spoke of allowing themselves permission to self-progress intensity over time. 

“It gave me the confidence to push a little bit harder, knowing that they can see if I’m overdoing it”, Nancy 70yrs.

However, a few commented that they were uncomfortable engaging in MVPA and unlikely to continue regular physical activity at this intensity.

Self-monitoring of intensity was challenging for some. Participants received instruction in how to judge intensity and self-monitor heart rate; however, many wanted visual cues with equipment-based feedback (e.g., heart rate monitor for home use).

“I needed a better measure of exertion than just feeling puffed. Like monitor the heart rate to give back feedback in saying this is what it should feel like at a particular heart rate at home”, Frank 62 yrs.

Finding out what worked for them was another essential aspect of experiencing MPVA. Participants were encouraged to try different types of equipment and various activities to inform their choice moving forward. Rest between sets of the activity was a common coping strategy for those faced with stroke-related fatigue.

“Because my stroke did not cause physical damage only the face, I get tired very quickly after the stroke, but I used to have a break every now and then, just a minute or two breaks and then was ok”, George 67 yrs.

Convenience

Participants spoke about how convenience influenced their ability to participate. For most participants, convenience described factors associated with accessibility and the opportunity to do it themselves in a comfortable environment. The centrally located community-based gym within a local sporting facility in the Newcastle metropolitan area was easily accessible for many. A few participants who lived far from the center said they needed to schedule other activities, such as shopping or doctor appointments, to make travelling the distance worthwhile. Session scheduling was a concern for some who believed that a lack of time, distance to travel, returning to employment, and inflexibility in the scheduling of sessions made it difficult for them to commit to the twice-weekly sessions.

“Being self-employed I have to take the work as it comes, the timing was not good for me, especially as I have to travel 100k’s kilometers to get there”, Chris 52yrs.

Going by themselves was an aspect of convenience that offered participants the opportunity to use the gym facilities other than during the program. Most participants thought that familiarity with the gym setting and gym staff contributed to feelings of safety when engaging in independent MVPA. Family members were encouraged to accompany the participants both during the program and during independent attendance.

“The whole family have decided to go to the gym. My wife goes so I’ve been very lucky in that way. I don’t exercise at home I go to the gym, I tried to do exercise at home but there’s always things to do, like you say I’ll just do this I’ll just do that and whereas when you go to the gym there’s no excuses”, Joe 63yrs.

A comfortable environment and availability of equipment were essential considerations. For some, the environment was comfortable with a good choice of equipment and friendly staff.

“They have so many people in the place who are just there to help you…They come over and start talking to you”, Dean 42yrs.

However, for others, being in the gym environment was not as desirable.

“I didn’t like the gym…it doesn’t equate to me to be in that gym…I never did like the male-female gym environment anyway… I feel very intimidated”, Sheila 65 yrs.

### 3.4. What I Got out of It

#### 3.4.1. Making Changes

Overall, most participants reported making changes to their lifestyle behaviours. All participants expressed a commitment to making changes in moving more during the day. However, not all participants were committed to continuing similar physical activity levels as was experienced during the program.

“I’m just trying to find out what works for me in my new world so to speak”, Scott 54yrs.

Thinking differently

“Thinking differently”, a subcategory of “Making changes”, includes finding value in participation.

“It was made clear that diet and exercise were important so I could see the value in it all and got tougher on myself”, Roger 76yrs.

Some participants were excited about losing weight, increasing fitness, and taking control of their way forward, and described this as connecting with “my why”. Doing it for the family was another critical aspect that shaped individual motivation for maintaining health, as many were concerned about becoming a burden for their family resulting from a subsequent catastrophic stroke. 

“I’ve got dreams and goals so I know what I want to do and where I want to go so now I want to get there and so being unhealthy is not going to help me do it”, Chris 52yrs.

Some attributed feelings of personal accountability and used strategies such as making goals or diary scheduling to maintain motivation. However, many were concerned they would not have the internal drive to exercise after the program ended.

“I shall have to push myself on some occasions, and the difficult part will be to make sure that I continue over that amount of time, and not let other things get in the way, the choice is mine and I’m going to have make sure that I do it. I’ve been given the motivation; I’ve been given the tools, now I have to use them”, Anne 76yrs.

Doing it myself

“Doing it myself” is another subcategory of “Making changes”. Most participants self-reported regular physical activity on at least an additional three days to the two scheduled program sessions.

Participants used various strategies to continue regular physical activity, including planning regular daily physical activity, having a family member to exercise with, and enrolling in other paid activities (e.g., golf, aqua aerobics, or gym membership).

“I work hard enough, I went to the gym to do weights…, I did the boxing class…, I did aqua-aerobics, in the water I can really push myself, so I mean I try hard”, Chris 52yrs. 

Using a personal fitness device (e.g., Fitbit or smartwatch) during independent physical activity was a strategy used to self-monitor progress by some participants. Visualizing progress gave them a sense of achievement and pride in accomplishing new targets.

“It gave me the accomplishments... like I’ve now walked 1.2 kilometers…. or I’ve lifted five kilos. I just felt good and then I’d like to give myself a tick. Well I’ve done that today. So I think it’s a… little self-accomplishment tick”, Mena 46yrs.

Walking was the most commonly reported activity. However, some participants tried different activities to find what worked for them (e.g., swimming or aqua aerobics) with pain the driver for finding alternative modes. 

A small proportion of the women believed that housework was ample to meet their needs as incidental physical activity.

“I don’t do anything. We’ve got a big house and I was doing the cleaning and doing the gardening outside”, Mary 75yrs.

#### 3.4.2. Feeling Better

“Feeling better” is the second category of “What I got out of it” and includes participant experiences associated with a sense of fitness and psychological factors. 

“It was a very good experience. I felt better while I was doing it, I felt better than I did before. I felt good in myself and about myself", Sam 77yrs.

Sense of fitness

“Sense of fitness” is a subcategory of the experience of feeling better. For some, fitness was associated with a sense of energy. Although participants reported being tired after their sessions, they generally spoke of increasing fitness and increased energy levels over time.

“I didn’t have enough energy to do things, I’d try and do something and I’d go for 5 minutes and have to sit down…I’m over that, I’m strong enough to keep going now”, John 86yrs,

One participant reported how engaging in MVPA after work helped him relax. However, many who had returned to the workforce reported little energy to commit to physical activity after work. Some tried to maintain a daily routine, but others found they could not cope with their fatigue.

“After working I’m exhausted the next day, it takes me nearly 2 days to get over it”, Bessie 62 yrs. 

“Getting stronger” is another subcategory that participants believed contributed to a sense of fitness, with some expressing surprise at their ability to do more than first perceived. 

“It worked for me ….my overall strength has increased and improved, so I think it has been worth the effort”, Anne 76yrs.

Participants with stroke-related balance issues attributed improvements to getting stronger. 

“My walking is a lot better; I’m not looking like I’m drunk anymore”, Mena 46yrs.

Increased muscle strength and a sense of fitness helped some participants sustain regular physical activity.

“I think that the exercise has definitely helped. I’m stronger than I was before, and I think that because I’m fitter it takes less energy to do things” Doug 63yrs.

Psychological factors

For some, physical activity had a calming effect on their anxiety levels that facilitated emotional stability; this allowed them to feel more relaxed, sleep better and generally feeling well in and about themselves. 

“I know what an anxiety attack is like, and I know what post-traumatic stress is like, it’s cruel. I know for my body, anxiety and emotional stress, I need exercises to burn that stress out”, Terry 57yrs. 

Many participants believed participation in the program gave them confidence, explaining that they were unsure of their physical capabilities following their stroke event.

“I feel that it’s given me permission… I feel that I’m now allowed to do that, rather than holding back and being a bit precious. I have more confidence”, Jill 59yrs.

Factors that facilitate gaining confidence include the supervised and independent practice of MPVA, health professional guidance, discussion, and peer support.

“It let me see that I could do things that I didn’t think I could do. I didn’t think I’d be ever able to do the things in the gym, which was probably a ridiculous notion on my part but that was how I felt”, Margaret 76yrs.

Many participants said that they looked forward to the intervention sessions, enjoyed the feelings associated with engaging in physical activity, and enjoyed the companionship of others. Enjoyment commonly motivated sustained physical activity. 

“I had something to look forward to. I was happier because of the interaction you know and I looked forward to going every Tuesday and Thursday”, Rose 74yrs.

## 4. Discussion

This study offers an insight into participant experiences of a secondary stroke prevention program held within a community-based gym for people with TIA or mild stroke. Data analysis provided two concepts. The first concept, “What it offered me”, describes critical elements that shape participant experience of the program (health professional support, being in a group and meeting my needs). The second concept, “What I got out of it”, describes perceived benefits of program participation (making changes and feeling better). Our findings highlight aspects of secondary stroke prevention programs that people with TIA and mild stroke perceive acceptable for guiding them in reducing their risk of recurrent stroke. 

Recent systematic reviews have identified limited evidence for effective secondary stroke programs that support people with TIA and mild stroke population to increase time spent engaging in MVPA [32]. Furthermore, existing lifestyle interventions for secondary stroke prevention are insufficient at influencing the physical activity behaviour of this patient population [18]. Evidence in preparation by the authors suggests that most people who have experienced TIA and mild stroke make no change to their physical activity behaviour post-stroke event, irrespective of previous levels of physical activity [33], adding gravity to the importance of delivering secondary stroke prevention programs that have strong client acceptability.

Participants perceived that receiving in-person health professional support was pivotal to enable ‘hands-on’ demonstration and support, provided credible knowledge, and tailoring the intervention to accommodate for individual challenges and needs. Participants, most of whom had little to no significant physical impairments limiting their ability to engage in the recommended level of physical activity, expressed a strong desire for health professional support. This desire suggests that factors other than physical ability may influence physical activity behaviour changes (e.g., anxiety, lack of awareness) among people with TIA and mild stroke [34,35,36]. Previous evaluations of secondary stroke interventions suggest that people with TIA and mild stroke want inclusion in the development of—and prescription of—physical activity [37]; regular health professional follow up [37]; simplified written information for self-directed physical activity [20]; and signposting for community-based physical activity [20]. While our study reflects similar findings, study participants emphasised the ability to practise MVPA within a safe environment with trusted feedback provided by a qualified health professional. Our findings align with two systematic reviews investigating factors that contribute to the successful delivery of physical activity and lifestyle interventions that highlight the importance of health professional support in providing activity demonstration, biofeedback, practise, and graded tasks [10,38]. 

“Thinking differently” and “Doing it myself” are two essential aspects of making changes that reflect the participant’s move towards internalising motivation for engaging in MVPA. A move towards self-determined motivation from a position of external motivation is a goal of programs underpinned by the self-determination theory. Self-determination theory is a well-established meta-theory that commonly forms the framework for behaviour change interventions and programs supporting the maintenance of behaviour over time [39]. Participant perceptions of what the program offered them (i.e., in the concept of “What it offered me”) resonated with the three basic psychological needs identified in the self-determination theory: (a) competence, (b) autonomy, and (c) relatedness [39].

Seeing value in program participation and experiencing positive emotions (e.g., enjoyment of participation, delight in learning new skills) are essential factors that participants perceived facilitated their willingness to make changes [40,41]. Changes made by the participant to their physical activity behaviour resonates with the organismic integration theory, a sub-theory within the self-determination theory [41]. This theory proposes a continuum of motivation from a position derived solely from external input to self-determined intrinsic motivation derived from enjoyment by performing the activity [40,41].

The opportunity to practice MVPA helped study participants achieve the confidence and competence necessary for independent MVPA. Initially, participants reported engaging in MVPA to be challenging. However, most participants found that they could achieve set benchmarks and self-progress individualised routines. The positive changes in participant physical activity behaviour reflect the basic physiological needs theory, also a sub-theory within the self-determination theory framework [42]. The basic psychological needs theory postulates that increases in self-esteem and higher levels of engagement in physical activity are associated with a positive relationship between satisfying the three basic psychological needs of competence, autonomy, and relatedness, and a movement towards greater self-determination of motivation [42]. Furthermore, competence is strengthened with an achievable challenge, receiving in-person encouragement and acknowledgement of competence, and the opportunity to initiate self-practice (e.g., attending gym independent of program schedule) [39,43]. Additionally, perceived competence is a strong predictor of behaviour adherence and is associated with the individual’s sense of autonomy [40].

Although feelings of competence strongly relate to the ability to make behaviour changes [39,44], volitional behaviour, and the necessity for the individual to see value or reasoning behind making changes to behaviours, are equally as important [40,43] in supporting autonomy or the perception of personal empowerment and freedom of choice [45,46]. Study participants placed value on making choices for themselves, being included in program suitability and being in a safe environment. However, some participants expressed a desire for a more systematic approach to identifying individual needs.

Study participant perceptions of health professional empathy, approachability, and tone of conversation echoes evidence highlighting the importance of provider support (i.e., eliciting feelings that “my health professional likes me”) and non-judgmental, non-controlling communication [40,47]. This perceived need for feelings of warmth, empathy, and acknowledgement of emotions is akin to relatedness [46,48]. Relatedness (feelings of mutual support and being valued by others) is closely linked to a greater likelihood that individuals will internalise, adopt, and sustain behaviours if they have a sense of connection with the person providing support [42]. Our findings support evidence that shows autonomy and needs satisfaction are stronger in combined health professional-led and group interventions [49] as participants expressed that coming together in a health professional facilitated group setting allowed them the opportunity to share experiences with others who had experienced similar life disruptions. Furthermore, our findings echo benefits associated with group-based interventions described by Yolam et al. [50], which include instilling and maintaining hope, strengthening identity with other group members (Universality), promoting peer-to-peer relationships (altruism), and mutual support [50].

The strength of this study lies in the number of participants (n = 30) willing to share their experiences. The large number of interviews (n = 51) collected over a prolonged period and the large number of participants who consented to interviews at both time points (n = 20) allowed for richness and breadth of categories derived from the data to portray the fullness of participants’ experiences and strengthen dependability. A rich description of participant characteristics with diversity in age, sex, and time since stroke event, and concise description of study context, strengthened transferability. However, our study is limited as study participants self-selected to participate in SLAM-TIA, were from a program delivered at a single regional site, and had limited ethnicity, cultural and language diversity. As reporting of participant perceptions lacks long-term follow up, we cannot determine factors that influence ongoing physical activity compliance.

### Implications for Policy and Practice

This study highlights important clinical implications when translating evidence into ‘real world’ health service practice. 

Consistent face-to-face health professional interaction is critical for guidance, feedback, and support;A group setting offers the opportunity to share experiences, model exercise behaviour and offer relatedness through peer support;Tailoring interventions is necessary to meet individual needs and foster autonomy.

## 5. Conclusions

Physical inactivity is a significant risk factor of recurrent stroke, increasing physical activity at guideline recommendations is an essential component of recurrent stroke risk reduction. Programs to reduce the risk of recurrent stroke must be feasible, safe, and acceptable to those who have had a TIA or mild stroke. This study suggests that people with TIA or mild stroke who participated in a community-gym-based health service delivered program perceived value in the accompanying health professional support and being in a group, highlighting the role these features play in changing physical activity behaviours. 

## Figures and Tables

**Figure 1 ijerph-18-11448-f001:**
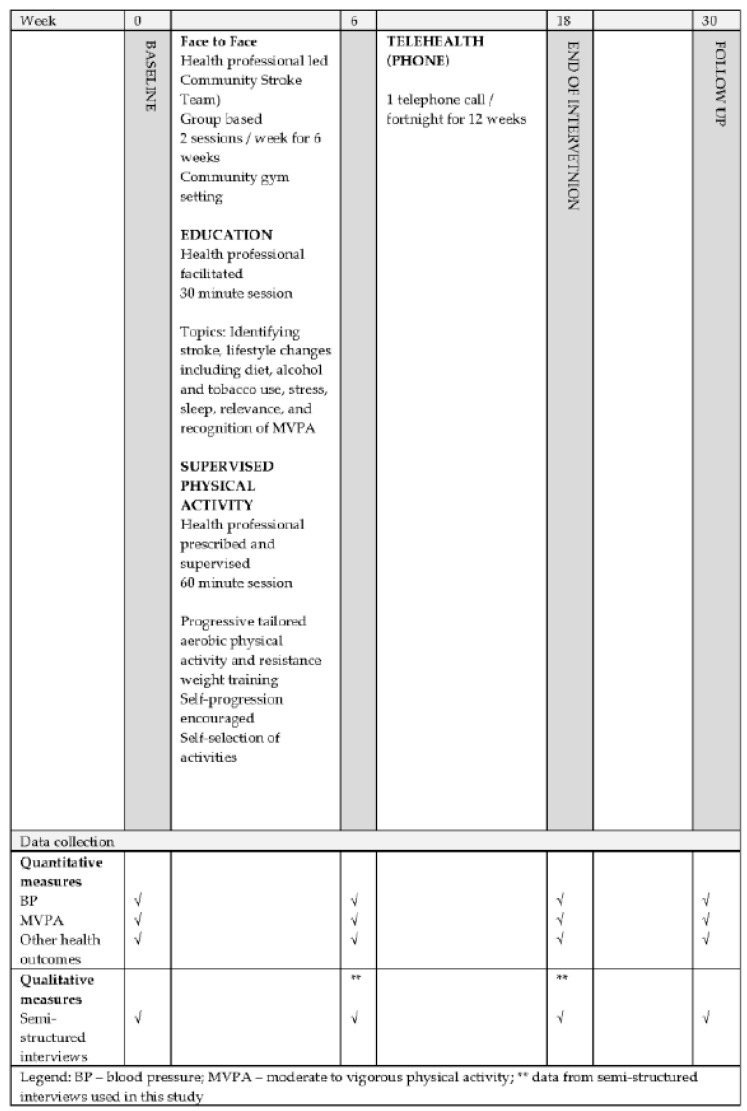
Timeline of secondary stroke prevention program and data collection.

**Table 1 ijerph-18-11448-t001:** Characteristics of 30 participants interviewed for qualitative study investigating experiences of participating in a secondary stroke prevention program.

Characteristic	Participants (n = 30)
Age (*years), mean (SD), range*	66 (10), 42 to 86
≤65, n (%)	15 (50)
≥66 n (%)	15 (50)
Sex, n (%)	
Female	13 (43)
BMI, (kg/m^2^), n (%)	
Normal (≤24.9)	8 (27)
Overweight (25–29.5)	10 (33)
Obese (≥ 30)	12 (40)
Diagnosis, n (%)	
TIA	10 (33)
Ischemic stroke	16 (54)
Haemorrhagic stroke	4 (13)
Time since stroke event *(days), mean (SD) n=28*	160 (137)
Fatigue Assessment score at baseline, n (%)	
Fatigue (24–34)	9 (30)
Severe fatigue (≥ 35)	2 (7)
Depression, Anxiety and Stress Scale (DASS) score ≥ 7, n (%)	
Cumulative total	21 (70)
Depression	12 (40)
Anxiety	13 (43)
Stress	17 (57)

**Table 2 ijerph-18-11448-t002:** Framework for the concepts, ‘What it offered me’ and ‘What I got out of it’ with supporting categories and subcategories.

CONCEPT	CATEGORY	SUBCATEGORY
WHAT IT OFFERED ME	Health Professional support	In-person Helpful Information Telehealth-based health coaching
	Being in a group	Mutual support Exercising with others
	Meeting my needs	Experiencing MVPA Convenience
WHAT I GOT OUT OF IT	Making changes	Thinking differently Doing it myself
	Feeling better	Sense of fitness Psychological factors

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
