# Peer review of "Participants’ Perspective of Engaging in a Gym-Based Health Service Delivered Secondary Stroke Prevention Program after TIA or Mild Stroke"

_ijerph, 2021, doi:10.3390/ijerph182111448_

Round 1

Reviewer 1 Report

The authors presented a descriptive study of  'Participants' perspective of engaging in a gym-based health  service delivered secondary stroke prevention program after 3 TIA or mild stroke.'

When I started to read the paper, I hoped the authors would gather more general conclusions from their collected data. Such assumptions could allow them to construct a more generalized model of factors influencing engaging stroke patients into gym-based rehabilitation.

Unfortunately, the authors settle for less level of generalization, which is only a descriptive method. The authors have put a lot of work into preparing the paper, especially for the time-consuming interviews; however, they were presented descriptively.

In such a study, I would expect more quantitative data. The two models the authors created are also based on choosing patients' opinions what makes it less objective and could not allow us to rule out some dose of personal choosing (even if it is subconscious).

Other issues and shortcomings

-- In the first sentence (40-41), the authors stated that 

 "A transient ischemic attack (TIA) is a focal and rapid neurological disturbance resulting from cerebral vascular occlusion…."

It needs explanation or rewriting because TIA resulting from focal cerebral ischemia is not associated with permanent cerebral infarction (Eaton 2009).

So the occlusion is unnecessary for TIA development, and even, to say more, prolonged occlusion of the vessel will cause complete stroke, but not  TIA. Therefore authors should state that occlusion must be transient; another cause of TIA could be any vascular disease (unnecessary occlusion)  leading to focal neurological deficits within the brain.

-- Although the work is descriptive, the Authors delivered some numbers; unfortunately, they seemed to be not precise. In table 2 and line 179, the authors say that they examined thirty patients, although, in the same table 2, readers could find numbers as below;

 Age (years),

66 to 76, n (%)   15 (46)

< 65, n (%)   12 (36)

> 77, n (%)   6 (18)

It makes together 33 participants (15+12+6), and if we look for Depression, Anxiety and Stress Scale (DASS), we found 34 participants (however in this category some patients could overlap within two types – if so it should be explained in a table or in the table's footnotes). If we consider the BMI category, we found 33 patients and,  in the diagnosis section 31. This discrepancy does not strengthen readers' confidence in the reliability of the tests.

It is  this is undoubtedly interesting work in some aspects,

However, in my opinion, the paper's scientific level and the topic do not match the scientific profile of the IJERPH journal.

Author Response

October 21st, 2021

Dear Dr Freene,

Subject: Manuscript ID: ijerph 1416716 “Participants’ Perspective of Engaging in a Gym-based Health Service Delivered Secondary Stroke Prevention Program after TIA or Mild Stroke”

We wish to thank you for your time and for offering the opportunity to revise our manuscript.

Please find our responses to the comments by the reviewers regarding our manuscript.

We hope that the explanations and alterations assist in clarifying any uncertainties.

Yours sincerely,

Maria Sammut

PhD candidate
School of Health Sciences
Faculty of Health and Medicine
M: +61 451 122 963
E: c3217252@uon,edu.au

The University of Newcastle (UON)
University Drive
Callaghan NSW 2308
Australia

CRICOS Provider 00109J

Response to Reviewers

Thank you for reviewing and commenting on our manuscript.

We offer our explanations for changes made to the manuscript in the following table:

Reviewer comment

Author response

Reviewer 1

Thank you for your comments.

When I started to read the paper, I hoped the authors would gather more general conclusions from their collected data. Such assumptions could allow them to construct a more generalized model of factors influencing engaging stroke patients into gym-based rehabilitation.

This paper reports on a qualitative study, carried out using qualitative descriptive methodology. The qualitative descriptive approach was deemed the most appropriate to answer the research question, based on a review of existing research and research methodology literature.

Generalisability, the extension of research findings from a study on a sample population to a population at large, applies only to quantitative methods and required data on large populations. As such, it is not an appropriate standard to apply to this study.

The principle of transferability is more applicable to qualitative research and small-scale studies. Transferability is the process whereby the reader judges the applicability of the finding of the study to his or her own situation. It is strengthened by the authors’ provision of a description of participant characteristics and the study context, as was provided in this paper.

Qualitative description is widely used in health research to offer a rich description of patient experiences and attitudes of identified phenomena (1), in this case the participating in a secondary stroke prevention program. Its value in bringing about change and improvement in health service is well recognised. (2-4). Furthermore, Chafe (2017) describes the powerful role of qualitative description’s purposely-low inference analysis in providing a readily understandable synopsis of the research outcomes (2). Sandelowski (2000) explains that choosing qualitative description allows the researcher to “seek descriptive validity, or an accurate accounting of events that most people (including researchers and participants) observing the same event would agree is accurate, and interpretive validity, or an accurate accounting of the meanings participants attributed to those events that those participants would agree is accurate”(4).

In such a study, I would expect more quantitative data

The study reported in this paper is a qualitative study, and by definition does not contain quantitative data other than for the characteristics of study participants.

Qualitative approaches are identified as the most appropriate option when the research aims centre on hearing participants’ voices and understanding a process from their perspective.

Unfortunately, the authors settle for less level of generalization, which is only a descriptive method. The authors have put a lot of work into preparing the paper, especially for the time-consuming interviews; however, they were presented descriptively

As identified in our response to comment number one, generalisability is only applicable to quantitative research and can only be achieved with data from large populations. As such, it is not an appropriate criterion by which to judge the value of a qualitative study such as this.

Study transferability is one component of study trustworthiness. The case for transferability is strengthen when discussing study strengths and limitations.

Page 16 line 604

“A rich description of participant characteristics with diversity in age, sex and time since stroke event and concise description of study context adds to transferability. However, our study is limited as study participants self-selected to participate in SLAM-TIA, were from a program delivered at a single regional site; and had limited ethnicity, cultural and language diversity”.

The two models the authors created are also based on choosing patients' opinions what makes it less objective and could not allow us to rule out some dose of personal choosing (even if it is subconscious).

Rather than choosing opinions, the process involved systematic approaches to sampling and data analysis consistent with the overarching methodology. The value and expectation of ‘objectivity’ is consistent with the positivist ontology that underpins quantitative research and rests on a view that there is a singular ‘reality’. This study is informed by the constructivist paradigm and as such a relativist ontology that views realities as multiple, contextual and socially constructed.

The quality of qualitative research is judged by different criteria than quantitative research. While objectivity is not the target, there are arrange of strategies employed in this study to ensure it meets the criteria of credibility, dependability and confirmability. These are outlined in the trustworthiness section of the article’s methods.

Page 7 line 174-186

Many mechanisms were used to support study trustworthiness [(5)]. Member checking was not possible because of potential bias in the overarching trial. However, regular oral, analytical discussion, and peer critique ensured strong logic and data linkage to emergent coding and categories to strength study credibility [(5)]. Analysis of all interviews ensured a wide range of participant perceptions and various experiences. Furthermore, input from experienced researchers within the field of stroke and qualitative research contributed to the quality of analysis to ensure data interpretation occurred in context. Journaling study logistics, using a standardised interview guide, maintaining a methods log, and detailed memo writing to augment contemplation and provide consistency between data and findings strengthened study dependability and audibility. Ongoing scrutiny of personal biases, experience and background influences (Reflexivity) continued throughout the analysis process to reflect on any researcher preconceptions that may influence the interpretation of participant voice [(6)]”.

-- In the first sentence (40-41), the authors stated that 

 "A transient ischemic attack (TIA) is a focal and rapid neurological disturbance resulting from cerebral vascular occlusion…."

It needs explanation or rewriting because TIA resulting from focal cerebral ischemia is not associated with permanent cerebral infarction (Eaton 2009).

So the occlusion is unnecessary for TIA development, and even, to say more, prolonged occlusion of the vessel will cause complete stroke, but not  TIA. Therefore authors should state that occlusion must be transient; another cause of TIA could be any vascular disease (unnecessary occlusion)  leading to focal neurological deficits within the brain

Thank you for your input. We have changed the description of our study population to increase clarity.

We have altered the previous statement, page 2 line 40-45

“A transient ischemic attack (TIA) is a focal and rapid neurological disturbance resulting from cerebral vascular occlusion with no residual infarction and complete resolution of symptoms [(7, 8)]. Although focal cerebral vascular occlusion associated with mild stroke (NIHSS ≤ 4 mRS 0-1) results in residual infarction [(9)], people with mild stroke generally experience rapidly resolving symptoms and are assessed as having minimal to no residual physical deficits [(9, 10) ]

For improved reader understanding:

Page 2 line 40 – 45

“A transient ischemic attack (TIA) is a focal and rapid neurological disturbance resulting from a temporary disruption of atrial blood flow with no residual brain tissue death (infarction) and complete resolution of symptoms. (8, 11) While persistent disruption of blood flow results in brain tissue infarction and subsequent stroke (9, 10), people with mild stroke generally experience rapidly resolving symptoms and are assessed as having minimal to no residual physical deficits(9)”.

Although the work is descriptive, the Authors delivered some numbers; unfortunately, they seemed to be not precise. In table 2 and line 179, the authors say that they examined thirty patients, although, in the same table 2, readers could find numbers as below;

 Age (years),

66 to 76, n (%)   15 (46)

< 65, n (%)   12 (36)

> 77, n (%)   6 (18)

It makes together 33 participants (15+12+6), and if we look for Depression, Anxiety and Stress Scale (DASS), we found 34 participants (however in this category some patients could overlap within two types – if so it should be explained in a table or in the table's footnotes). If we consider the BMI category, we found 33 patients and,  in the diagnosis section 31. This discrepancy does not strengthen readers' confidence in the reliability of the tests.

Thank you for bringing this to our attention this error it has been corrected

Page 6 line 119

Table 2. Characteristics of 30 participants interviewed for qualitative study investigating experiences of participating in a secondary stroke prevention program

Characteristic

Participants

(n=30)

Age (years), mean (SD), range

66 (10),
42 to 86

   ≤ 65, n (%)

15 (50)

   ≥ 66 n (%)

15 (50)

Sex, n (%)

Female

13 (43)

BMI, (kg/m2), n (%)

   Normal (≤ 24.9)

8 (27)

   Overweight (25-29.5)

10 (33)

   Obese (≥ 30)

12 (40)

Diagnosis, n (%)

   TIA

10 (33)

   Ischemic stroke

16 (54)

   Haemorrhagic stroke

  4 (13)

Time since stroke event (days), mean (SD) n=28

160 (137)

Fatigue Assessment score at baseline, n (%)

   Fatigue (24-34)

9 (30)

   Severe fatigue (≥ 35)

2 (7)

Depression, Anxiety and Stress Scale (DASS) score ≥ 7, n (%)

   Cumulative total

21 (70)

   Depression

12 (40)

   Anxiety

13 (43)

   Stress

17 (57)

However, in my opinion, the paper's scientific level and the topic do not match the scientific profile of the IJERPH

As identified in our responses to earlier comments, the methodology used in the study reported in this paper is widely used in health research and is recognised for its value in direct and richly described communication that allows stakeholders to relate to, discuss and act upon. (2)

Suitability of the methodological approach for publication in the International Journal of Environmental Research and Public Health is evidenced by the recent publication of papers using the same methodology, such:

Int. J. Environ. Res. Public Health 202017(23), 9150; https://doi.org/10.3390/ijerph17239150 

Realtors’ Perceptions of Social and Physical Neighborhood Characteristics Associated with Active Living: A Canadian Perspective Gavin R. McCormack 1,2,3,4,*, Autumn Nesdoly 5 , Dalia Ghoneim 1 and Tara-Leigh McHugh

  1. Neergaard MA, Olesen F, Andersen RS, Sondergaard J. Qualitative description – the poor cousin of health research? BMC Medical Research Methodology. 2009;9(1):52.
  2. Chafe R. The Value of Qualitative Description in Health Services and Policy Research. Healthc Policy. 2017;12(3):12-8.
  3. Maxwell JA. Understanding and validity in qualitative research. Harvard educational review. 1992;62(3):279-300.
  4. Sandelowski M. Whatever happened to qualitative description? Res Nurs Health. 2000;23(4):334-40.
  5. Lincoln Y, Guba E. Establishing Trustworthiness. Naturalistic Inquiry. Newbury Park, CA: Sage; 1985. p. 295-7.
  6. Charmaz K. Constructing Grounded Theory. 2nd Edition ed. D S, editor. London UK: SAGE Publications; 2014.
  7. Albers GW, Caplan LR, Easton JD, Fayad PB, Mohr JP, Saver JL, et al. Transient Ischemic Attack — Proposal for a New Definition. New England Journal of Medicine. 2002;347(21):1713-6.
  8. Easton JD, Saver JL, Albers GW, Alberts MJ, Chaturvedi S, Feldmann E, et al. Definition and Evaluation of Transient Ischemic Attack. Stroke. 2009;40(6):2276-93.
  9. Roberts PS, Krishnan S, Burns SP, Ouellette D, Pappadis MR. Inconsistent Classification of Mild Stroke and Implications on Health Services Delivery. Archives of Physical Medicine and Rehabilitation. 2020;101(7):1243-59.
  10. Schwartz JK, Capo-Lugo CE, Akinwuntan AE, Roberts P, Krishnan S, Belagaje SR, et al. Classification of Mild Stroke: A Mapping Review. PM&R. 2019;11(9):996-1003.
  11. Mendelson SJ, Prabhakaran S. Diagnosis and Management of Transient Ischemic Attack and Acute Ischemic Stroke: A Review. JAMA. 2021;325(11):1088-98.

Reviewer 2 Report

First of all, thank you for allowing me to review this interesting manuscriptt, whose objetive Is to explore perceptions of participation in a secondary stroke prevention program (delivered by a community-based multidisciplinary health service team within a community gym) by adults with TIA or mild stroke

The current manuscript is suitable for publication; only a minor revision of the language and style are advised

Reviewer 3 Report

  • The paper is well written and everything is comprehensible.
  • Table 2: The participants’ characteristics are the appropriate for such a study, while the number of the participants it seems appropriate, too.
  • Table 2: Authors should explain if it is ok to have statistically significant different number of participants with normal BMI as compared to the overweight and obese groups.
  • In general, this study consists of well organized methodologies that are fully supported by the exported results.

Round 2

Reviewer 1 Report

The authors improved the paper to some extent. The rest of the methodological comments are as before.